# Oxidative Stress, Glutathione Metabolism, and Liver Regeneration Pathways Are Activated in Hereditary Tyrosinemia Type 1 Mice upon Short-Term Nitisinone Discontinuation

**DOI:** 10.3390/genes12010003

**Published:** 2020-12-22

**Authors:** Haaike Colemonts-Vroninks, Jessie Neuckermans, Lionel Marcelis, Paul Claes, Steven Branson, Georges Casimir, Philippe Goyens, Geert A. Martens, Tamara Vanhaecke, Joery De Kock

**Affiliations:** 1Department of In Vitro Toxicology and Dermato-Cosmetology (IVTD), Vrije Universiteit Brussel (VUB), Laarbeeklaan 103, 1090 Brussels, Belgium; haaike.colemonts-vroninks@vub.be (H.C.-V.); Jessie.Neuckermans@vub.be (J.N.); Paul.Claes@vub.be (P.C.); Steven.Branson@vub.be (S.B.); Tamara.Vanhaecke@vub.be (T.V.); 2Laboratoire de Pédiatrie, Hôpital Universitaire des Enfants Reine Fabiola (HUDERF), Université Libre de Bruxelles (ULB), Avenue J.J. Crocq 1-3, 1020 Brussels, Belgium; Lionel.MARCELIS@huderf.be (L.M.); georges.casimir@huderf.be (G.C.); pgoyens@ulb.ac.be (P.G.); 3Department of Laboratory Medicine, AZ Delta General Hospital, Deltalaan 1, 8800 Roeselare, Belgium; Geert.Martens@azdelta.be; 4Center for Beta Cell Therapy in Diabetes, Vrije Universiteit Brussel (VUB), Laarbeeklaan 103, 1090 Brussels, Belgium

**Keywords:** hereditary liver disease, tyrosinemia type 1, nitisinone, transcriptomics, oxidative stress, glutathione metabolism, liver regeneration

## Abstract

Hereditary tyrosinemia type 1 (HT1) is an inherited condition in which the body is unable to break down the amino acid tyrosine due to mutations in the fumarylacetoacetate hydrolase (FAH) gene, coding for the final enzyme of the tyrosine degradation pathway. As a consequence, HT1 patients accumulate toxic tyrosine derivatives causing severe liver damage. Since its introduction, the drug nitisinone (NTBC) has offered a life-saving treatment that inhibits the upstream enzyme 4-hydroxyphenylpyruvate dioxygenase (HPD), thereby preventing production of downstream toxic metabolites. However, HT1 patients under NTBC therapy remain unable to degrade tyrosine. To control the disease and side-effects of the drug, HT1 patients need to take NTBC as an adjunct to a lifelong tyrosine and phenylalanine restricted diet. As a consequence of this strict therapeutic regime, drug compliance issues can arise with significant influence on patient health. In this study, we investigated the molecular impact of short-term NTBC therapy discontinuation on liver tissue of Fah-deficient mice. We found that after seven days of NTBC withdrawal, molecular pathways related to oxidative stress, glutathione metabolism, and liver regeneration were mostly affected. More specifically, NRF2-mediated oxidative stress response and several toxicological gene classes related to reactive oxygen species metabolism were significantly modulated. We observed that the expression of several key glutathione metabolism related genes including *Slc7a11* and *Ggt1* was highly increased after short-term NTBC therapy deprivation. This stress response was associated with the transcriptional activation of several markers of liver progenitor cells including *Atf3*, *Cyr61*, *Ddr1*, *Epcam*, *Elovl7*, and *Glis3*, indicating a concreted activation of liver regeneration early after NTBC withdrawal.

## 1. Introduction

Hereditary tyrosinemia type 1 (HT1, OMIM #276700) is the most prevalent and severe of the five tyrosine-related inherited metabolic disorders (TIMD). The overall incidence of this rare autosomal recessive disease is estimated at 1 in 100,000 newborns worldwide [1]. However, significantly larger numbers of new patients have been reported in specific regions including the province of Quebec, Canada and Northern Europe, i.e., Finland and Norway [2,3,4,5]. HT1 is characterized by a defective or absent fumarylacetoacetate hydrolase (FAH) enzyme, the final enzyme of the tyrosine degradation pathway that is predominantly active in liver and kidney cells (Figure 1) [6,7]. It manifests as a severe necrotic liver disease accompanied by hepatomegaly, cirrhosis, impaired coagulation, acute-on-chronic liver failure resulting in jaundice, ascites, bleeding and neurological crises [1,6]. This chronic liver injury likewise increases the risk for hepatocellular carcinoma (HCC) development. HT1 also affects normal kidney function causing renal tubular defects with rickets, Fanconi-like syndrome and glomerulosclerosis [8]. The damage to the liver and kidneys is caused by the accumulation of toxic upstream metabolites fumarylacetoacetate and maleylacetoacetate that cause protein denaturation (Figure 1) [1,9]. These toxic intermediates are downstream metabolized into toxic derivatives succinylacetoacetate and succinylacetone (SA) which play a major role in the onset of neurological crises (Figure 1) [10]. Without proper and early treatment, HT1 can present itself under three main clinical forms: (i) The acute form, which is associated with acute liver failure during the first months of life, failure to thrive and, in most cases, death of the patients in their infancy; (ii) the subacute form that usually appears within the second half of the first year after birth as hepatomegaly and rickets due to renal tubular defects; and (iii) the chronic form which presents after the first year of age and shows slower disease progression [11].

Historically, tyrosine- and phenylalanine-restricted diets have been applied as therapy for HT1 patients, but have proven to be inadequate, especially in case of acute and subacute clinical forms [12,13]. Orthotopic liver transplantation is being applied, but only for those patients that suffer from acute-on-chronic liver failure or that have developed HCC due to a delayed diagnosis as a result of inadequate neonatal screening programs [14,15,16]. For more than twenty years, standard therapy for HT1 patients consists of the lifelong daily intake of the drug NTBC (2-(2-nitro-4-trifluoromethylbenzoyl)-1,3-cyclohexanedione; nitisinone), a former herbicide and potent inhibitor of the key enzyme 4-hydroxyphenylpyruvate dioxygenase (HPD) [17,18]. NTBC therapy causes an effective upstream metabolic block that prevents the production of the aforementioned toxic metabolites (Figure 1) [2]. However, although NTBC therapy rescues HT1 patients from severe illness and early death, it also leads to tyrosine accumulation in the blood [19], the production of alternative tyrosine-derivatives including N-acetyltyrosine [20], the disturbance of monoamine neurotransmitter metabolism [21,22], and the symbiotic perturbation of host and microbiome tryptophan metabolism [23]. Noteworthy changes are also present in the ratio of several amino acids suggesting that the availability of amino acids for neurotransmitter biosynthesis and liver function may be altered [22]. Due to the aforementioned effects of NTBC therapy, concerns exist for patients to develop ocular, cutaneous and possible neurological complications including mood changes, lower IQ and depression as observed in hereditary tyrosinemia type 2 (HT2,OMIM #276600) and type 3 (HT3, OMIM #276710) patients as a consequence of hypertyrosinemia [24,25,26]. Corneal keratopathy, with the risk of developing blindness, has been detected even with low dose NTBC in TIMD patients [27]. NTBC therapy, due to the concomitant hypertyrosinemia, requires therefore lifelong dietary adjustment by a tyrosine- and phenylalanine-restricted diet in order to reduce the debilitating side-effects of the drug. Unfortunately, due to this strict therapeutic regime, drug compliance issues could become a problem for HT1 patients with significant impact on their health.

To gain fundamental knowledge on the impact of non-drug compliance, we investigated which molecular mechanisms and pathways become modulated in the liver as soon as the inhibitory effects of the drug NTBC become less effective. To accomplish this, we performed gene expression profiling and biochemical analyses of Fah-deficient mouse livers under continuous NTBC therapy and after seven days of NTBC therapy discontinuation.

## 2. Materials and Methods

### 2.1. Tyrosinemia Type 1 Mouse Model

The FRG mouse model (C57Bl/6J background; Yecuris) is characterized by a triple knockout of Fah^−/−^, Rag2^−/−^, and Il2rg^−/−^ and thus serves as a suitable model for HT1. Neonates die from acute liver failure if NTBC is not continuously administered [28]. Therefore, unless specified otherwise, all mice received continuous NTBC treatment (16 mg/L) through their drinking water, supplemented with 3% (*w*/*v*) dextrose for taste. Mice were fed with irradiated LabDiet^®^ 5LJ5 chow (LabDiet, St. Louis, MO, United States) ad libitum, which is a low-tyrosine and low-phenylalanine diet that acts as a surrogate for the protein-restricted diet of HT1 patients.

### 2.2. Animal Experiments

All mouse caring, breeding and experimental procedures were performed in accordance with the ethical standards of the Vrije Universiteit Brussel and National Rules on Animal Experimentation and were approved under grant numbers 15-210-2 and 15-210-4 by the local Ethical Committee for Animal Experiments of the Vrije Universiteit Brussel in Belgium. Mice were group-housed in individual ventilated cages (IVC) in a temperature (19–23 °C) and humidity (30–70% relative humidity) regulated environment with a 14/10-h light/dark cycle under continuous NTBC treatment (16 mg/L) administered through their drinking water. Cage environment was enriched with cardboard shelters, nesting material and wooden sticks. To the best of our abilities, results were reported in accordance with the ARRIVE guidelines [29]. At the age of 11 weeks, NTBC therapy was discontinued for seven consecutive days for the FRG-7dNTBC group whereas it was administered continuously for the FRG+NTBC group.

### 2.3. Sample Collection and Preparation

At the age of 12 weeks, all mice were anesthetized by intraperitoneal injection of ketamine/xylazine mixture (87.5 mg/kg Ketamidor^®^ (Ecuphar BV, Breda, The Netherlands)/12.5 mg/kg Rompun^®^ (Bayer Animal Health GmbH, Leverkusen, Germany)). Blood samples were collected by a ventral heart puncture with a 26Gx^1/2”^ needle (0.45 × 13 mm) and 1 mL syringe (Terumo^®^; VWR, Leuven, Belgium). Between 0.5–1 mL of blood was collected in ethylenediaminetetraacetic acid (EDTA) coated micro tubes (Sarstedt, Nümbrecht, Germany, K3E tube) together with 1–2 drops of blood on dried blood spot (DBS) cards (Whatman, Overijse, Belgium, 903; GE Healthcare). DBS cards were allowed to dry at room temperature for at least 24 h before use. Blood samples were centrifuged (1500× *g*, 15 min and 4 °C) and serum was frozen at −80 °C until further use. DBS cards were analyzed as described below within 1 month post collection. Liver tissue was collected and rinsed with ice cold physiological serum to remove remaining debris. Liver tissue cubes of approximately 1–2 cm^3^ were fixed at 4 °C in 4% (*w*/*v*) paraformaldehyde (Sigma-Aldrich, Overijse, Belgium) for 24 h, dehydrated in ethanol (VWR, Leuven, Belgium) series (using Microm GmbH STP 120-1, Prosan, Arnhem, The Netherlands) and embedded in paraffin (using Microm EC 350-1, Prosan) for histopathological analyses. For protein and transcriptome analysis, liver tissue cubes of max. 1 cm^3^ were collected in RNAprotect Tissue Reagent (Qiagen Benelux, Venlo, The Netherlands) and stored at −80 °C.

### 2.4. Western Blot

Frozen liver tissue samples were homogenized in a radio-immunoprecipitation assay buffer containing 1% (*w*/*v*) protease and phosphatase inhibitor cocktail, and 1% (*w*/*v*) EDTA 0.5 M (all from Thermo Scientific, Merelbeke, Belgium). Samples were sonicated for 30 s, rotated for 15 min at 4 °C, centrifuged for 5 min at 14,000× *g* and the supernatants were transferred into 2 mL microtubes (VWR, Leuven, Belgium). Protein quantification was performed on the supernatants using the Pierce™ bicinchoninic acid protein assay (Thermo Scientific) with bovine serum albumin (BSA) as reference standard. Both proteins (Fah and Hpd) were separated by electrophoresis on 12% (*w*/*v*) Mini-*Protein^®^* TGX Stain-Free™ precast gels (Bio-Rad, Temse, Belgium) and blotted onto nitrocellulose Trans-Blot^®^ Turbo™ Transfer packs (Bio-Rad). Membranes were blocked in blocking buffer consisting 5% (*v*/*v*) non-fatty milk in Tris-buffered saline solution (20 mM Tris and 135 mM sodium chloride) and 0.1% (*v*/*v*) Tween-20, followed by incubation overnight at 4 °C with primary antibody (both diluted 1:1000 in blocking buffer) directed against Fah (HPA041370) and Hpd (HPA038322) and several wash steps in Tris-buffered saline solution containing 0.1% (*v*/*v*) Tween-20 to remove excessive antibody (all from Sigma-Aldrich). Subsequently, membranes were incubated for 1 h at room temperature with polyclonal goat anti-rabbit secondary antibody (Dako, Agilent Technologies, Heverlee, Belgium) diluted 1:1000 in blocking buffer and washed again several times. Detection of the proteins was carried out by means of a Pierce™ enhanced chemiluminescence Western blotting substrate kit (Thermo Scientific) according to the manufacturer’s instructions. Chemiluminescent detection of both target proteins was done on a ChemiDoc™ MP Imaging System and analyzed using Image Lab 5.0 software (all from Bio-Rad, Temse, Belgium).

### 2.5. Isolation of RNA and Reverse Transcriptase-Polymerase Chain Reaction (PCR)

Total RNA was extracted from all samples using the GenElute Mammalian Total RNA Purification Miniprep Kit (Sigma-Aldrich) according to the manufacturer’s instructions. The isolated RNA was quantified at 260 nm using a Nanodrop spectrophotometer (Thermo Scientific, Merelbeke, Belgium). Total RNA was reverse transcribed into cDNA using iScript^TM^ cDNA Synthesis Kit (Bio-Rad) followed by cDNA purification with the Genelute PCR clean up kit (Sigma-Aldrich).

### 2.6. Quantitative Real-Time PCR (qPCR)

A StepOne Plus system (Thermo Scientific) was used for RT-qPCR using TaqMan fast advanced master mix and gene expression assays (see Appendix A). Data normalization was done against the mean of the reference genes glyceraldehyde-3-phosphate dehydrogenase (*Gapdh*), hydroxymethylbilane synthase (*Hmbs*) and ubiquitin C (*Ubc*) using qbase+ software (Biogazelle, Zwijnaarde, Belgium).

### 2.7. Dried Blood SPOT (DBS) Analysis

Used reagents and solvents (methanol, acetonitrile and formic acid) of the purest available quality (gradient HPLC) were supplied by Merck. AB Sciex API 3200 and API 4000 (both MS analyzers), coupled to an HPLC 1100 Agilent system were used for analysis. For the quantification of the amino acids tyrosine (m/z 238 > 102) and phenylalanine (*m*/*z* 222 > 102), discs of 3.2 mm diameter were punched, using a DBS Puncher (Perkin Elmer, Mechelen, Belgium), and placed into a 96-wellplate. Internal standards were added in separate wells and DBS were eluted for 20 min at room temperature using 200 µL methanol. 40 µL of the eluent was transferred into a second 96-wellplate, while the residues of the first plate were dried and stored for SA sampling (see hereafter). To each well of the second plate, 160 µL stock solution containing labeled amino acids standards (Cambridge Isotopes Labeled Amino Acid Standards NSK-A) was added. The plate was dried at 55 °C under nitrogen atmosphere for 15 min, reconstituted in 25 µL Butanol-HCl mixture (Sigma-Aldrich) and incubated for 20 min at 65 °C under inert atmosphere. After final evaporation under a flow of nitrogen, 200 µL of eluting solution (acetonitrile–water–formic acid mixture (80:20:0.5)) was added. The samples were analyzed by MS/MS through direct flow injection using the same eluting solution. Final concentrations were obtained using labeled internal standards (^13^C_6_ Phenylalanine, *m*/*z* 228 > 102 and ^13^C_6_ Tyrosine, *m*/*z* 244 > 102). For SA-quantification (*m*/*z* 211 > 137), the aforementioned dried samples (plate 1) were for 40 min at 50 °C derivatized by 200 µL hydrazine hydrate solution (acetonitrile-water-formic acid-hydrazine mixture (80:20:0.5:0.5) with deuterated SA as internal standard (*m*/*z* 216 > 142)). Samples were transferred to a new 96-wellplate to discard punches and subsequentially analyzed by MS/MS through direct flow injection. Final concentrations were obtained using a standard curve. For NTBC measurements (*m*/*z* 330 > 218 and 330 > 126), discs of 3.2 mm diameter were punched using a cutting tool (Sigma–Aldrich) and placed into 1.5 mL Eppendorf tubes. Punches were eluted for 30 min at room temperature using 200 µL of pure methanol and 1 µM of internal standard (^13^C_6_ nitisinone, *m*/*z* 336 > 218 and 336 > 126). After discarding the punch, solutions were directly used in LC/MS. 10µL of sample solution was eluted by a Poroshell 120 EC-C18 column (Agilent) using a isocratic method (0.5 mL/min acetonitrile–water mixture (85:15) with 0.05% formic acid) at 30 °C. Final concentrations were obtained using a standard curve.

### 2.8. Histopathology

Paraffin-embedded liver tissue was deparaffinized using xylene (VWR) and decreasing concentrations of ethanol (VWR). Next, a nuclear counterstaining was performed using Mayer’s hematoxylin (Sigma-Aldrich). Subsequently, samples were dehydrated using increasing concentrations of ethanol and xylene and mounted with acrytol (Leica, Diegem, Belgium).

### 2.9. Microarray Profiling

Total RNA (100 ng) was used for amplification and in vitro transcription using the Genechip 3’ IVT Express Kit following the manufacturer’s instructions. The amplified RNA (aRNA) was purified with magnetic beads and 15 μg Biotin-aRNA was treated with fragmentation reagent. 12.5 μg fragmented aRNA was hybridized to Affymetrix Mouse 430 2.0 arrays along with a hybridization cocktail solution and then placed in a Genechip^®^ Hybridization Oven-645 rotating at 60 rpm at 45 °C for 16 h. After incubation, arrays were washed on a Genechip^®^ Fluidics Station 450 and stained with the Affymetrix HWS kit as indicated by the manufacturer’s protocols. The chips were scanned with an Affymetrix GeneChip^®^ Scanner 3000 7G and the quality control matrices were confirmed with the Affymetrix GCOS software following the manufacturer’s guidelines. Background correction, summarization (median polish) and normalization (quantile) were done with Robust Multiarray Analysis (all from Affymetrix, Merelbeke, Belgium). The data discussed in this publication have been deposited in the NCBI Gene Expression Omnibus and is accessible through GEO Series accession number GSE161478. Heatmaps and volcano plots were generated using the Transcriptome Analysis Console (TAC) version 4.0 (Thermo Scientific). Ingenuity Pathway Analysis software (version 2020) was used for gene set enrichment and pathway analyses considering only genes with a ≥ 5-fold difference in expression and False Discovery Rate (FDR) *p*-value ≤ 0.05.

## 3. Results

### 3.1. Short-Term NTBC Therapy Discontinuation Inflicts Severe Liver Damage in Fah-Deficient Mice 

To study the impact of short-term NTBC therapy discontinuation in HT1 patients, FRG mice deficient in Fah were used as surrogate model. Hence, in a first set of experiments, we confirmed the knockout of *Fah* in FRG mouse livers by western blot analysis (Figure 2A) as well as the constitutive expression of *Hpd*, the therapeutic target of NTBC, in these livers (Figure 2B). Wildtype (WT) C57Bl/6 mice were used as positive controls (Figure 2A,B).

Next, NTBC therapy of FRG mice was discontinued for seven consecutive days (FRG-7dNTBC) or continuously administered (FRG+NTBC) as shown in Figure 2C. We found that albumin (*Alb*) expression was slightly, but not significantly, decreased whereas alpha-fetoprotein (*Afp*) was significantly increased (245-fold) in Fah-deficient mouse livers after short-term NTBC discontinuation (65.14 ± 50.17%) versus continuous NTBC administration (94.34 ± 48.91%) (Figure 2D,E). Quantification of NTBC blood levels in FRG mice at seven days post withdrawal confirmed the drug therapy discontinuation (0.25 ± 0.08 µM) versus FRG mice under continuous NTBC treatment (1.73 ± 0.22 µM) (Figure 2F). Furthermore, daily follow up of HT1 blood parameters showed that SA levels raised steadily whereas tyrosine and phenylalanine levels remained quite stable after NTBC therapy withdrawal (Figure 2G–I). Finally, histopathological investigations confirmed the presence of severe liver lesions in NTBC-deprived FRG mice as opposed to FRG mice under continuous NTBC therapy (Figure 2J,K). More specifically, FRG mice continuously treated with NTBC only presented mild hepatocellular changes as hepatocytes were enlarged and regularly showed large dysmorphic nuclei (N) with several well-formed nucleoli (Figure 2J; Appendix A). In strong contrast, FRG mice at seven days post withdrawal of NTBC presented severe hepatocellular changes including necrosis of hepatocytes, enlarged hepatocytes, large dysmorphic nuclei (N), oval-like cells (black arrow), and infiltration of small lymphoid cells with dark nuclei and scant cytoplasm (yellow arrow) (Figure 2K; Appendix A).

### 3.2. Glutathione Metabolism and Oxidative Stress Related Pathways Are Activated in Fah-Deficient Livers upon Short-Term NTBC Therapy Discontinuation

Total RNA, extracted from liver tissue of FRG+NTBC and FRG-7dNTBC mice, was subjected to whole transcriptome profiling using Affymetrix Mouse 430 2.0 arrays and Ingenuity Pathway Analysis software. We found that seven days after NTBC therapy discontinuation, 4.65% (2096 genes) of the expressed genes was more than 2-fold and 0.72% (324 genes) even more than 5-fold up regulated, whereas 3.45% (1555 genes) of the expressed genes was significantly down regulated for more than 2-fold and 0.60% (271 genes) was even more than 5-fold decreased (Figure 3A,B; Appendix A).

When considering only genes that were at least 5-fold up or down regulated after short-term NTBC therapy discontinuation, we found that these genes grouped together in toxicological gene classes related to liver disease (*Fibrosis of liver*, *Progressive familial intrahepatic cholestasis type 1*, *Intrahepatic cholestasis*, *Hepatic steatosis*, *Cholestasis*), liver damage (*Hepatic injury*, *Apoptosis of liver cells*, *Liver damage*, *Cell death of hepatocytes*, *Cell death of liver cells*, *Necrosis of liver*), liver regeneration (*Proliferation of liver cells*, *Liver regeneration*) and liver cancer (*Metastatic liver tumor*, *Hepatocellular carcinoma*) of which *Hepatocellular carcinoma* was the most pronounced toxicological gene class harboring 62 significantly modulated genes including alpha-fetoprotein (*Afp*) that was 24-fold increased (Figure 3C,D; Appendix A). Furthermore, upstream regulator analyses support these findings (Appendix A). More specifically, in the context of liver damage and regeneration, we found that the hepatogenic growth factors epidermal growth factor (*Egf*) and hepatocyte growth factor (*Hgf*) were predicted to be active as well as the pro-fibrotic transforming growth factor bèta 1 (*Tgfb1*) (activation z-score ≥ 2; Appendix A).

Canonical pathway analyses showed that pathways related to glutathione metabolism (*Glutathione-mediated detoxification*, *Glutathione redox reactions I*) as well as oxidative stress response (*NRF2-mediated oxidative stress response*) were activated in the liver seven days after NTBC therapy withdrawal (Figure 3E). In contrast, the *Aryl hydrocarbon receptor signaling* and *LXR/RXR activation* pathways were found to be inhibited (Figure 3E). With respect to the *NRF2-mediated oxidative stress response* pathway, 118 genes were found to be at least 5-fold up regulated, whereas 58 genes were found to be at least 5-fold down regulated (Figure 3F). More specifically, *Abcc4* (11-fold), *Cbr1* (7-fold), *Gsta1* (8-fold), *Gpx2* (13-fold), *Gsr* (13-fold), *Hacd3* (8-fold), *Hmox1* (59-fold), *Hspb8* (6-fold), *Mgst2* (40-fold), *Mgst3* (23-fold), *Nqo1* (15-fold), and *Txnrd1* (5-fold) were found to be among the top increased genes after short-term NTBC therapy discontinuation related to the *NRF2-mediated oxidative stress response* (Figure 4C). In addition, several gene classes related to reactive oxygen species (ROS) metabolism were found to be significantly modulated (Figure 4A,B). As such, we found that the ROS-related genes *Adm*, *Bcl2l11*, *Cdkn1a*, *Cyp1a1*, *G6pd2*, *G6pdx*, *Ggt1*, *Gmfb*, *Gsr*, *Hmox1*, *Lcn2*, *Maoa*, *Mt2*, *Nqo1*, *Pgd*, *Pon3*, *Ptgs2*, *Slc7a11,* and *Ucp2* were at least 5-fold up regulated whereas the genes *Car3*, *Egfr*, *Nox4*, *Prodh,* and *Tert* were at least 5-fold down regulated (Figure 4D). Out of these ROS-related genes the top increased genes were *Slc7a11* (300-fold) and *Ggt1* (121-fold) (Figure 4D,F,G).

In the context of glutathione metabolism, respectively 13 genes were at least 5-fold up and 7 genes down regulated in the *Glutathione redox reactions I* pathway whereas for the *Glutathione-mediated detoxification* pathway 17 genes were at least 5-fold up regulated and 8 genes at least 5-fold down regulated (Figure 3F). Moreover, the expression of the following glutathione metabolism related genes *Mgst2* (40-fold), *Mgst3* (23-fold), *Gstm2* (15-fold), *Gstm5* (7-fold), *Gpx2* (13-fold), *Gsr* (13-fold), and *Gsta1* (8-fold) was found to be highly increased after short-term NTBC therapy deprivation (Figure 4E).

### 3.3. Fah-Deficient Liver Tissue Exhibits Increased Expression of Liver Progenitor Markers after Short-Term NTBC Therapy Discontinuation

*Liver regeneration* was one of the major toxicological gene classes that was found to be significantly modulated in Fah-deficient mouse livers when NTBC therapy was discontinued (Figure 3C,D). Therefore, we investigated the gene expression of a large set of quiescent and activated liver progenitor cell (LPC) and biliary epithelial cell (BEC) markers. These LPCs and BECs are endogenous cells that are known to differentiate into hepatocytes and cholangiocytes, and frequently accompany chronic liver diseases (Figure 5A) [30]. In this context, we found that the gene expression of the LPC/BEC markers *Atf3* (19-fold), *Cyr61* (7-fold), *Ddr1* (6-fold), *Elovl7* (38-fold), *Glis3* (8-fold), and *Hspa1a* (6-fold) was significantly increased more than 5-fold in Fah-deficient livers when NTBC therapy was discontinued for seven consecutive days using microarray analyses (Figure 5A,B).

Hierarchical clustering using differentially-expressed LPC/BEC markers allowed to discriminate between liver tissue of Fah-deficient mice under continuous NTBC therapy and after seven days of NTBC therapy withdrawal (Figure 5C). Furthermore, RT-qPCR analyses of commonly used LPC/BEC markers confirmed that *Atf3* (83-fold), *Cyr61* (5.5-fold), *Ddr1* (12-fold), *Elovl7* (266-fold), and *Glis3* (13-fold) gene expression was significantly increased and showed that *Epcam* expression was increased approximately 4-fold in Fah-deficient livers when NTBC therapy has been discontinued (Figure 6). In contrast, the observed increase in *Hspa1a* expression could not be confirmed using RT-qPCR and no significant difference in gene expression was observed for other commonly used LPC markers *Cd24a*, *Krt19,* and *Prom1* (Figure 6).

## 4. Discussion

Since its introduction, NTBC therapy has clearly improved the vital prognosis and the quality of life of HT1 patients. However, due to its strict drug and associated dietary regime, compliance problems have been described [31,32,33]. More specifically, it has been reported that HT1 patients have presented with recurrent porphyria-like neurological crises after discontinuation/interruption of NTBC treatment. For some of the patients, these crises were life-threatening and accompanied by respiratory muscle paralysis requiring ventilator support, hemodynamic disturbance, acute progressive ascending motor neuropathy causing profound impairment, recurrent seizures, and neuropathic pain [10]. To gain insight on the molecular impact of non-drug compliance by HT1 patients, we compared transcriptome profiles of liver tissue of Fah-deficient mice under continuous NTBC therapy and after short-term, i.e., seven days NTBC therapy discontinuation.

Biochemical blood analyses of Fah-deficient mice after NTBC withdrawal showed increased SA levels, indicating that the inhibitory effects of the drug NTBC became inefficient seven days after the last dose which was associated with liver damage. SA can also be considered as an indirect indicator of redox imbalance as it has been shown that treatment of rodents with SA methyl ester leads to 5-aminolevulinic acid (ALA) accumulation that has been associated with induced oxidative subcellular and tissue damage within the liver [34]. This is in accordance with previous findings in similar HT1 mouse models showing that the half-life of NTBC in mouse plasma was 54 h. Hpd is therefore only completely inhibited during the first three days following a single-dose of NTBC and fully regains its enzymatic activity seven days after NTBC therapy discontinuation [35]. Furthermore, transcriptome analyses of liver tissue revealed that molecular pathways associated with oxidative stress, glutathione metabolism and liver regeneration become activated in HT1 mice upon short-term NTBC therapy discontinuation.

With respect to genes involved in oxidative stress, ROS formation and glutathione metabolism, *Ggt1* (121-fold) and *Slc7a11* (300-fold) expression was mostly increased. *Ggt1* encodes an enzyme that catalyzes the transfer of the glutamyl moiety of glutathione (GSH) to a variety of amino acids and dipeptide acceptors. The observed highly increased gene expression of *Ggt1* in Fah-deficient liver tissue after short-term NTBC deprivation is in accordance with high gamma glutamyl transferase activity levels observed in untreated HT1 patients [36]. *Slc7a11* encodes the cystine/glutamate antiporter (xCT) and imports cystine as a precursor for GSH synthesis that supports antioxidant responses [37]. This massive increase in *Slc7a11* expression indicates that GSH depletion is already present seven days post NTBC withdrawal. This might be explained by the accumulation of the powerful GSH depletor fumarylacetoacetate, which triggers secondary mechanisms such as xCT to compensate for the shortage of GSH [9]. Recently, it was also reported that *Slc7a11* is a key player in liver regeneration as its overexpression in hepatocytes enhances, and its suppression inhibits, liver repopulation following toxic injury [38]. Therefore, the increased expression of *Slc7a11* cannot only be considered as an indication that GSH metabolism becomes activated in the liver after short-term NTBC therapy discontinuation to counteract ROS formation. It also suggests that liver regeneration processes have started [38].

To support this, we compared the LPC/BEC marker expression profile of liver tissue derived from NTBC-treated Fah-deficient mice and those that were deprived from NTBC for seven days. This gene expression profile was previously established using meta-analyses of human and mouse biliary epithelial cell gene profiles [30] and allowed us to discriminate between the two experimental groups. More specifically, we found that the gene expression of several LPC/BEC markers including *Atf3*, *Cyr61*, *Ddr1*, *Elovl7*, *Epcam*, *Glis3*, and *Hspa1a* was highly increased in Fah-deficient livers when NTBC therapy was discontinued for seven consecutive days. This data could be confirmed by RT-qPCR analyses for all markers except *Hspa1a*.

*Atf3*, also known as liver regeneration factor 1, belongs to the basic leucine zipper family of transcription factors [39]. It is an early response gene that is rapidly induced upon partial hepatectomy and induced in hepatocyte cultures stimulated with mitogenic growth factors [39,40]. A large number of studies have shown that *Atf3* is a stress-inducible gene [41], but may also prevent apoptosis [42], stimulate cell proliferation [43], and tumor invasion [44].

*Cyr61* encodes a matricellular protein that is involved in dampening and resolving liver fibrosis, mediates cholangiocyte proliferation and the ductular reaction. Both are repair responses to cholestatic injury of the liver [45]. 

*Ddr1* belongs to a unique family of receptor tyrosine kinases containing a discoidin homology domain in their extracellular region. DDRs are expressed during early embryonic development in different tissues of which *Ddr1* is mainly expressed in epithelia [46]. Interestingly, *Ddr1* expression has been reported in hepatocytes and bile duct epithelial cells of cirrhotic liver, but its role remains largely unclear [47].

*Elovl7* encodes a key enzyme involved in the elongation of very long chain fatty acids and has been shown to be up regulated upon hepatocyte dedifferentiation [48].

*Epcam* is a surface epithelial marker that is selectively expressed in the biliary tree, where it discriminates liver progenitor cells from mature cholangiocytes [49]. In addition, mature hepatocytes within the hepatic parenchyma also do not express *Epcam* [49]. Moreover, it is a commonly used marker to isolate LPCs from liver disease samples [50]. Furthermore, it was previously reported that Epcam^+^/Afp^+^ hepatocellular carcinoma subtypes exhibit features of hepatic stem/progenitor cells and display hepatic cancer stem cell-like traits including the abilities to self-renew and differentiate [51]. Therefore, our observed increased *Epcam* and *Afp* expression indicates that short-term NTBC therapy discontinuation might already be sufficient to put HT1 patients back at risk of developing hepatocellular carcinoma.

*Glis3* is a transcription factor containing a five Kruppel-like zinc finger motif. Its expression occurs early in embryogenesis and is thought to play a critical role in the cellular regulation of development by functioning as a repressor or activator of transcription. Mutations in the *Glis3* gene have led to neonatal diabetes, thyroid and renal diseases, and liver dysfunction ranging from hepatitis to cirrhosis [52].

*Hspa1a*, also known as Hsp70, has been shown to function as a chaperone during periods of cellular stress and induces the expression of several inflammatory cytokines identified as key players during early liver regeneration. Indeed, previous studies in mice have shown that the early phase of successful liver regeneration requires the presence of *Hspa1a* [53].

The massive up regulation of these LPC/BEC markers correlates with our histopathological observation of hepatocyte loss and oval-like cell proliferation and supports the predicted activation of liver repair mechanisms by our transcriptomics-based pathway analyses.

## 5. Conclusions

This study provides new data on the alteration of the HT1 liver transcriptome when NTBC therapy is withdrawn for a short period. We successfully identified a number of genes related to oxidative stress responses, glutathione metabolism and liver regeneration that were highly differentially-expressed versus continuous NTBC treatment. Especially in the context of liver regeneration we found a highly increased expression of a number of LPC/BEC markers when NTBC therapy was discontinued of which some, including *Epcam*, could also play a key role in hepatocellular carcinoma development associated with HT1 disease progression. However, to fully understand the pathophysiological impact of the experimentally observed transcriptional activation of oxidative stress, glutathione metabolism, and liver regeneration pathways upon short-term NTBC therapy discontinuation, combining these transcriptional data with subsequent integrative metabolomics studies will be of key importance [54,55]. High resolution metabolite profiling of, e.g., blood, urine, or cerebrospinal fluid will allow to study the direct and indirect consequences of the observed transcriptional changes in HT1 liver tissue at a biochemical level by identifying and measuring thousands of chemical features. In addition, experiments on HT1 patient-derived liver cells in vitro and in vivo in immune deficient mice after xenografting, will allow to further extrapolate our new findings from preclinical mouse studies to the situation in man [56].

## Figures and Tables

**Figure 1 genes-12-00003-f001:**
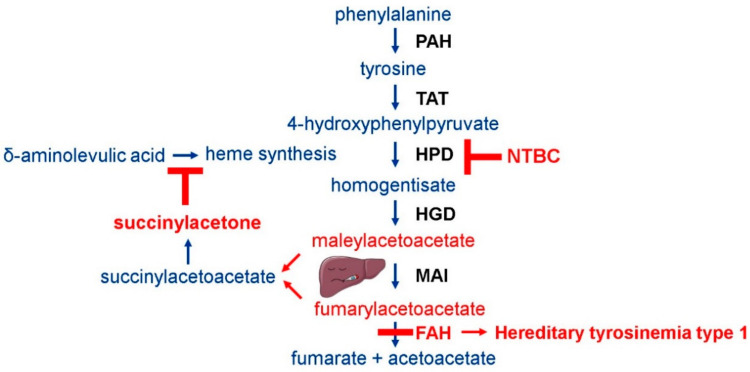
Tyrosine degradation pathway in liver cells. Loss of function of the fumarylacetoacetate hydrolase (FAH) enzyme causes the accumulation of toxic intermediate tyrosine metabolites maleyl- and fumarylacetoacetate, and subsequently, the production of succinylacetone through an alternative metabolization route. Nitisinone (NTBC) is a potent inhibitor of the upstream hydroxyphenylpyruvate dioxygenase (HPD) enzyme that prevents the formation of these toxic metabolites by providing a therapeutic block. Abbreviations: PAH, phenylalanine hydroxylase; TAT, tyrosine aminotransferase; HPD, 4- hydroxyphenylpyruvate dioxygenase; HGD, homogentisate dioxygenase; MAI, maleylacetoacetate isomerase; FAH, fumarylacetoacetate hydrolase, NTBC, nitisinone.

**Figure 2 genes-12-00003-f002:**
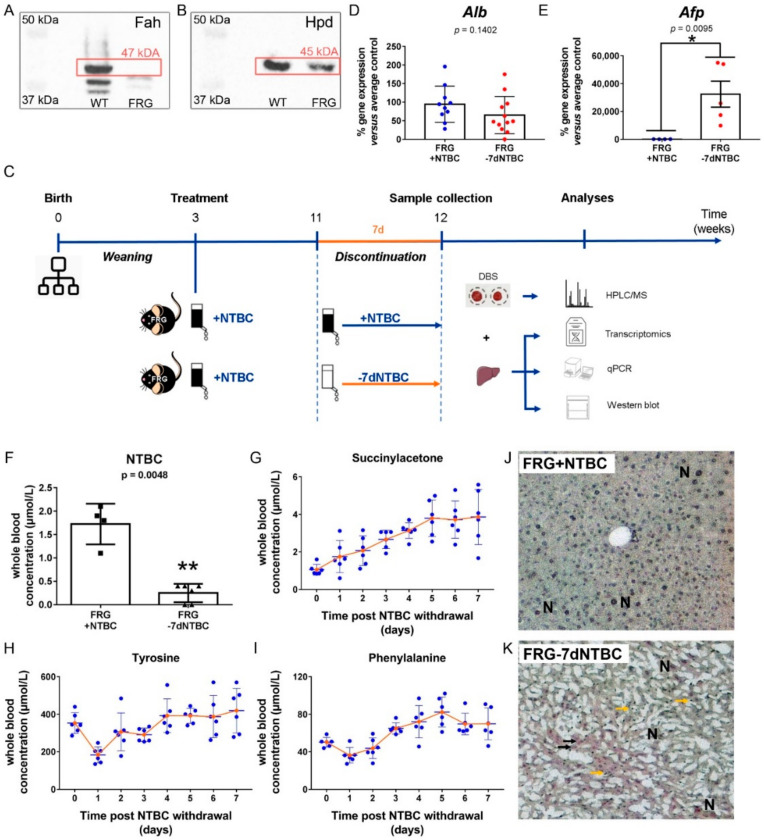
Molecular and biochemical evaluation of the FRG mouse model. FRG mouse livers are characterized by (**A**) the absence of Fah and (**B**) the presence of Hpd protein expression using western blot with healthy wildtype (WT) C57Bl/6 mice as positive controls. (**C**) Schematic representation of the NTBC drug therapy discontinuation experiment in FRG mice and subsequent sample collection. (**D**,**E**) Gene expression analysis of mouse albumin (*Alb*) and alpha-fetoprotein (*Afp*) by RT-qPCR. (**F**) Quantification of NTBC blood levels and daily follow up of (**G**) succinylacetone, (**H**) L-tyrosine and (**I**) L-phenylalanine levels using LC-MS/MS analyses of dried blood spots (DBS). Histopathological analysis of FRG mouse livers under (**J**) continuous NTBC therapy and (**K**) seven days post NTBC withdrawal by hematoxylin staining showing large dysmorphic nuclei (N), oval-like cells (black arrow) and small lymphoid cells (yellow arrow). * significantly increased; ** significantly decreased using a two-tailed Mann–Whitney test (*p* < 0.05).

**Figure 3 genes-12-00003-f003:**
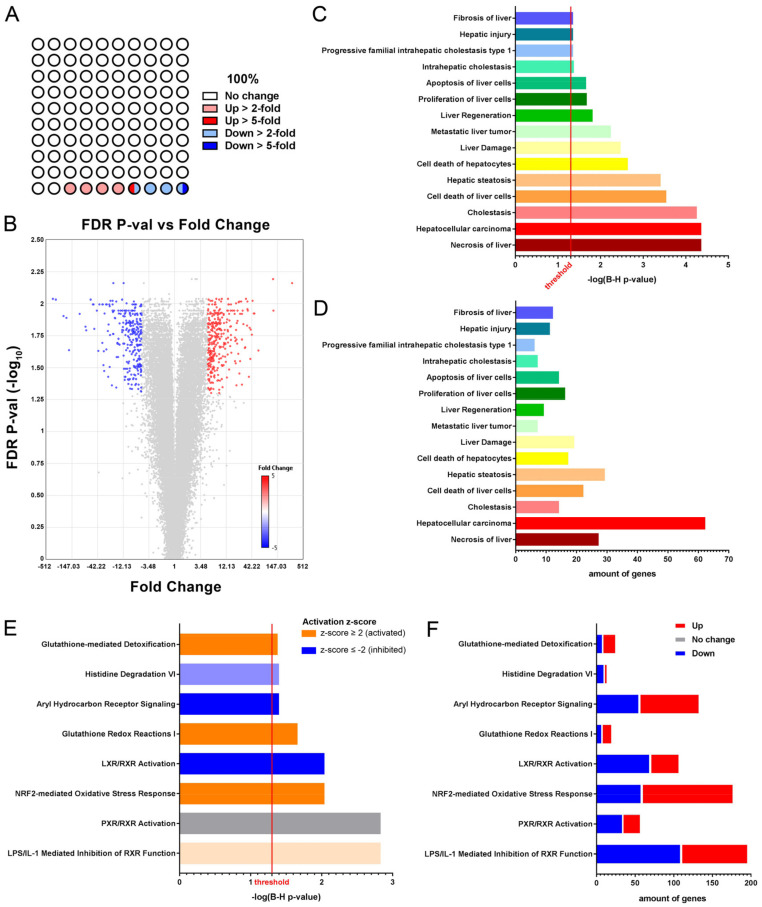
Whole transcriptome analyses of FRG liver tissue after seven days of NTBC therapy discontinuation versus continuous treatment. (**A**) Percentage of modulated genes in liver tissue. (**B**) Volcano plot displaying 5-fold up and down regulated genes with FDR *p*-value ≤ 0.05. (**C**,**D**) Toxicological gene class grouping of 5-fold up and down regulated genes showing the Benjamini–Hochberg (B–H) *p*-value of overlap and amount of modulated genes. (**E**,**F**) Canonical pathway analyses showing B–H *p*-value of overlap, activation z-scores and amount of modulated genes.

**Figure 4 genes-12-00003-f004:**
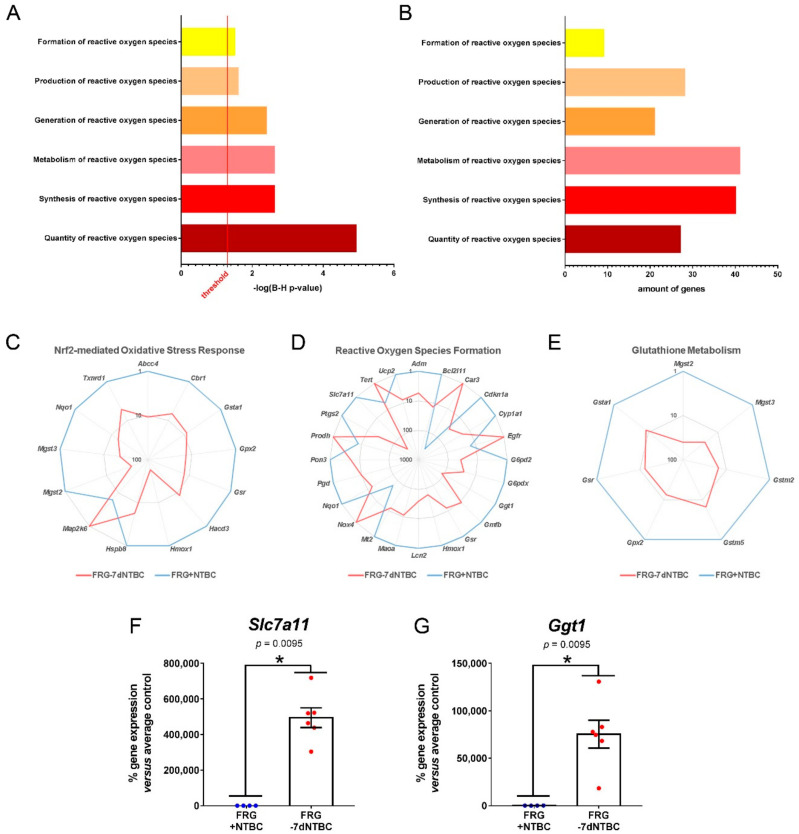
NTBC therapy discontinuation activates oxidative stress responses, reactive oxygen species (ROS) formation and glutathione metabolism. (**A**,**B**) ROS-related gene class grouping of 5-fold up and down regulated genes showing B–H *p*-value of overlap and amount of modulated genes. Spider graphs showing differentially-expressed genes related to (**C**) Nrf2-mediated oxidative stress response, (**D**) reactive oxygen species formation, and (**E**) glutathione metabolism. (**F**,**G**) Gene expression analysis of top increased ROS-related genes by RT-qPCR. * significantly increased using a two-tailed Mann–Whitney test (*p* < 0.05).

**Figure 5 genes-12-00003-f005:**
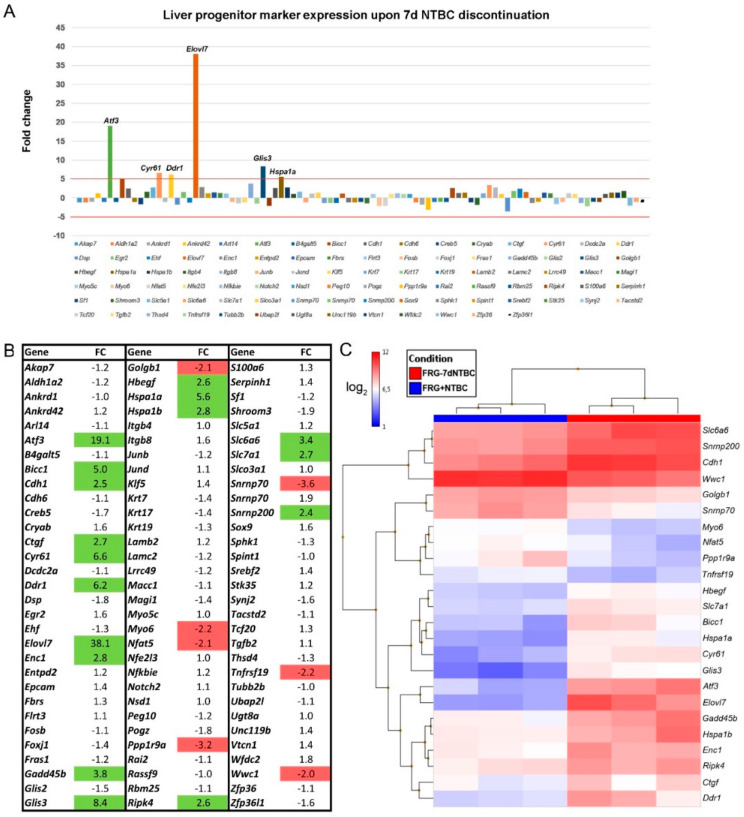
Liver progenitor cell (LPC)/biliary epithelial cell (BEC) marker expression in Fah-deficient mouse livers after short-term NTBC therapy discontinuation. (**A**,**B**) Gene expression profiling of LPC/BEC markers using microarray analyses. (**C**) Hierarchical clustering of differentially-expressed LPC/BEC markers.

**Figure 6 genes-12-00003-f006:**
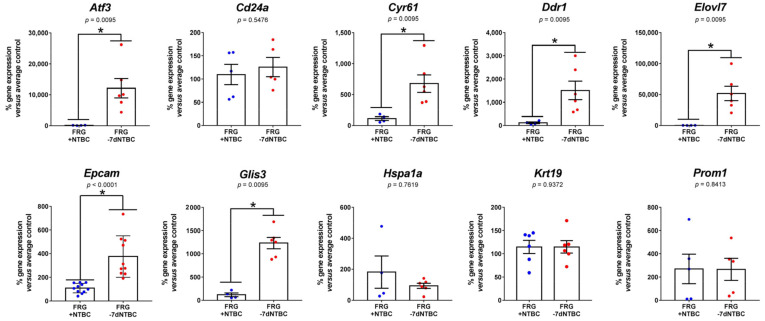
Gene expression analysis of LPC/BEC identification markers by RT-qPCR. * significantly increased using a two-tailed Mann–Whitney test (*p* < 0.05).

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
