# Peer review of "Oxidative Stress, Glutathione Metabolism, and Liver Regeneration Pathways Are Activated in Hereditary Tyrosinemia Type 1 Mice upon Short-Term Nitisinone Discontinuation"

_genes, 2020, doi:10.3390/genes12010003_

Round 1

Reviewer 1 Report

This paper consists of a metabolite-based transcriptome study on HT1 based on short-term NTBC withdrawal. Expression of Epcam and Afp was increased in the mouse model and linked to risk factors for hepatocellular carcinoma development.

This paper offers a fresh perspective on some biochemical pathways in the liver mediated by the drug presence and in order to improve this paper prior, I would suggest addressing/actioning the following specific comments:

  1. It is heartening to read a paper with so few grammatical/typographical errors and the authors should be commended on this, given the prevalence of such in the review sphere currently.
  2. The authors should mention specific alternative assessment methods to the work they have undertaken. Here I am thinking of metabolomics or associated approaches to the host/liver, or hepatic metabolic phenotyping work. Suggested references to include are highlighted in the final section of this review.
  3. The authors describe the methods remarkably, however in such studies, especially molecular mechanistic and mouse models, it is useful to have a section on the limitations of the approaches presented. This does not detract from the work but does indeed offer the reader with a perspective on future developments.
  4. Some of the figures are difficult to make out, such as Figure 3 where toxicological gene class groupings and CP analyses are difficult to make out.
  5. In Figure 4 - to what extent was the level of oxidative stress or ROS/RNS considered and what is the potential to analyse these? MS/MS and NMR methods can routinely be applied to analyse these from biofluids directly or alternatively by employing surrogates binding to the ROS/RNS.
  6. Is there a way to simplify Figure 5A, where each marker expression is included as part of the legend? This is difficult to read and perhaps including each of these within a table adjacent and with a designation may be helpful?

Suggested references as mentioned above:

  1. Wilmes, A., Limonciel, A., Aschauer, L., Moenks, K., Bielow, C., Leonard, M.O., Hamon, J., Carpi, D., Ruzek, S., Handler, A. and Schmal, O., 2013. Application of integrated transcriptomic, proteomic and metabolomic profiling for the delineation of mechanisms of drug induced cell stress. Journal of proteomics, 79, pp.180-194.
  2. Williams, I.H. and Wilson, P.B., 2017. SULISO: The Bath suite of vibrational characterization and isotope effect calculation software. SoftwareX, 6, pp.1-6.
  3. Lin, S., Qiao, N., Chen, H., Tang, Z., Han, Q., Mehmood, K., Fazlani, S.A., Hameed, S., Li, Y. and Zhang, H., 2020. Integration of transcriptomic and metabolomic data reveals metabolic pathway alteration in mouse spermatogonia with the effect of copper exposure. Chemosphere, p.126974.
  4. Waterer, G.W., 2012, June. Community-acquired pneumonia: genomics, epigenomics, transcriptomics, proteomics, and metabolomics. In Seminars in respiratory and critical care medicine (Vol. 33, No. 03, pp. 257-265). Thieme Medical Publishers.
  5. Grootveld, M., Percival, B., Gibson, M., Osman, Y., Edgar, M., Molinari, M., Mather, M.L., Casanova, F. and Wilson, P.B., 2019. Progress in low-field benchtop NMR spectroscopy in chemical and biochemical analysis. Analytica chimica acta, 1067, pp.11-30.
  6. Zhu, M., Li, L., Lu, T., Yoo, H., Zhu, J., Gopal, P., Wang, S.C., Porempka, M.R., Rich, N.E., Kagan, S. and Odewole, M., 2020. Uncovering biological factors that regulate hepatocellular carcinoma growth using patient derived xenograft assays. Hepatology

Reviewer 2 Report

--The manuscript would be improved by the addition of at least some qPCR results for key genes and by western blots of some of the key proteins suggested from the RNA results to have been altered.

--Additional time points would also have been welcomed, although we understand that additional k/o mice may not have been treated and samples prepared from them that would have allowed such results to be obtained.

--In Fig 2, I and J--the sizes and degree of magnification are too low really to be able to see what the changes were; It would be better if additional photomicrographs were provided. Additional description of the key changes by an experienced murine hepatopathologist would be a positive contribution, as well.

--In Table S2, why are multiple and somewhat different results presented for the same gene?  For examples, Slc7a11, Apf, Cdkh1a, Hspb1?
